# Overexpression of MicroRNA-138 Affects the Proliferation and Invasion of Urothelial Carcinoma Cells by Suppressing SOX9 Expression

**DOI:** 10.3390/biomedicines11113064

**Published:** 2023-11-15

**Authors:** Yuji Nitta, Tomomi Fujii, Tomoko Uchiyama, Aya Sugimoto, Takeshi Nishikawa, Maiko Takeda, Makito Miyake, Keiji Shimada, Kiyohide Fujimoto

**Affiliations:** 1Department of Diagnostic Pathology, Nara Medical University School of Medicine, Nara 634-8521, Japan; k140038@naramed-u.ac.jp (Y.N.);; 2Division of Fostering Required Medical Human Resources, Center for Infectious Disease Education and Research (CiDER), Osaka University, Osaka 565-0871, Japan; 3Department of Central Clinical Laboratory, Nara Medical University Hospital, Nara 634-8521, Japan; 4Department of Urology, Nara Medical University School of Medicine, Nara 634-8521, Japan; 5Department of Diagnostic Pathology, Nara City Hospital, Nara 630-8305, Japan

**Keywords:** urothelial carcinoma, SOX9, miR-138, autophagy, apoptosis

## Abstract

SRY-box transcription factor 9 (SOX9) is important for sexual differentiation, chondrogenic differentiation, and cell proliferation in cancer. It acts as a target molecule of microRNA (miR)-138 in various tumors and is associated with tumor development and growth. In this study, we analyzed the functions of miR-138 and SOX9 in urothelial carcinoma. SOX9 was highly expressed in invasive urothelial carcinoma tissues. miR-138 precursor transfection of T24 and UMUC2 cells significantly decreased SOX9 expression, indicating that SOX9 is a miR-138 target in urothelial carcinoma. Moreover, miR-138 precursor or SOX9 small interfering RNA (siRNA) transfection decreased the proliferation of urothelial carcinoma cell lines. To further confirm that miR-138–SOX9 signaling is involved in cell proliferation and invasion, urothelial carcinoma cells were transfected with the miR-138 precursor or SOX9 siRNA. This transfection reduced the proliferation and invasion of cells via the promotion of autophagy and apoptosis and G0/G1 cell cycle arrest. These results suggest that miR-138–SOX9 signaling modulates the growth and invasive potential of urothelial carcinoma cells.

## 1. Introduction

Urothelial carcinoma is the 10th most commonly diagnosed malignancy worldwide and is four times more common in men than in women [1]. Although the geographic and temporal patterns of urothelial carcinoma incidence are mostly based on tobacco smoking, other risk factors (occupational exposure to paint, rubber, aromatic amines, and other chemicals and arsenic contamination of drinking water) may be major causes in some populations [2,3,4,5]. Urothelial carcinomas are mostly non-invasive, with a good prognosis. This is partly due to the improvements in treatment modalities, such as endoscopic resection, adjuvant chemotherapy, and intravesical immunotherapy, which have decreased the mortality rate of this disease. In contrast, invasive urothelial carcinoma and carcinoma in situ can occur not only at de novo onset but also during the recurrence of non-invasive urothelial carcinoma. Urothelial carcinomas are solitary or multiple, whereas non-invasive urothelial carcinomas retain relatively original urothelial polarity and show low atypical morphology (low grade), and carcinoma in situ and most invasive carcinomas show highly irregular morphology (high grade) [6]. In invasive urothelial carcinoma, the tumor exhibits a desmoplastic stromal response and invades the stroma and vasculature. Multiple molecules may be involved in the acquisition of invasiveness by tumor cells [7,8].

MicroRNAs (miRNAs) are non-coding small RNAs consisting of approximately 20 bases that bind to the 3′-end of target molecules to induce various biological activities [9]. In various cancers, miRNAs disrupt the expression of various molecules, causing the autonomous proliferation, invasion, and migration of abnormal cells, leading to tumor growth and metastatic invasion [10,11]. Urothelial carcinomas are tumors with high differentiation potential, including glandular, squamous, and neuroendocrine differentiation [12,13]. In particular, highly atypical invasive urothelial carcinomas undergo various kinetic changes and differentiation under the action of miRNAs and their target molecules [14]. In this study, we investigated the expression of SRY-box transcription factor 9 (SOX9) in highly atypical invasive urothelial and intraepithelial carcinoma and identified miRNAs involved in cancer cell growth and invasion using SOX9 as a target molecule.

## 2. Materials and Methods

### 2.1. Cell Lines

This study was conducted on human carcinoma cell lines according to the ethical standards formulated in the Declaration of Helsinki. Two human urothelial carcinoma (UC) cell lines (T24 and UMUC2) were purchased from the American Type Culture Collection (Manassas, VA, USA). These cell lines were cultured in the Roswell Park Memorial Institute-1640 medium (Nakarai Tesque, Kyoto, Japan) supplemented with 10% fetal bovine serum and 50 U/mL penicillin–streptomycin (Nakarai Tesque, Kyoto, Japan). All cell lines were cultured at 37 °C and 5% CO_2_.

### 2.2. Tissue Samples and Immunohistochemistry

This study was approved by the ethics committee of Nara Medical University (NMU900, 3041), and informed consent was obtained from all patients. Formalin-fixed paraffin-embedded tissues surgically resected from 67 patients between 2018 and 2021 were used in this study. We examined 67 transurethral resections (TURs) or total cystectomies of UC specimens from patients who did not undergo chemotherapy or Bacillus Calmette–Guerin treatment (Table 1). Tumor stage and grade specimens were pathologically diagnosed and verified via visual inspection of hematoxylin and eosin-stained sections by at least two experienced pathologists (TF and TU). Sections were incubated with primary antibodies against SOX9, D2-40, and CD31 for 1 h at room temperature (22–25 °C). The reactions were visualized using a Histofine kit (Nichirei, Tokyo, Japan) with diaminobenzidine as the chromogen, followed by hematoxylin counterstaining.

### 2.3. miRNA Precursor and siRNA Transfection of UC Cell Lines

For transfection, three UC cell lines were seeded at a density of 1.5 × 10^5^ cells/well in a 6-well dish and transfected with 100 ng/L of siRNA against SOX9 for 72 h. Transfection with each siRNA and Ambion Pre-miR miRNA precursors (hsa-miR-138-5p, 675, 26a, 23b, 331-3p, 145, 27a, 345, 139, 197 and 367; Thermo Fisher Scientific, Waltham, MA, USA) was carried out using Lipofectamine RNAiMAX (Thermo Fisher Scientific), according to the manufacturer’s protocol. The following SOX9 siRNA (Hs_SOX9_1 FlexiTube siRNA; QIAGEN, Hilden, Germany) sequence was designed after selecting the appropriate DNA target sequences: 5′-ATGGGAGTAAACAATAGTCTA-3′.

### 2.4. Quantitative Reverse Transcription Polymerase Chain Reaction (qRT-PCR) Analyses of miRNA and mRNA

For the purification of total RNA, including miRNA from cells, we used the miRNeasy Mini Kit (QIAGEN, Hilden, Germany). For qRT-PCR, cDNA was synthesized from 1 μg of total RNA using the PrimeScript RT Master Mix (Perfect Real Time), SYBR Premix Ex Taq II, and Tli RNase H Plus (Takara, Otsu, Japan). qRT-PCR conditions were set at 95 °C for 30 s, followed by 55–63 °C for 30 s for a total of 35–45 cycles. The following PCR primers were used:

*SOX9* sense 5′-TCTCCTGGACCCCTTCATGA-3′,

*SOX9* antisense 5′-AACGTGTTCTCCTGGGGC-3′

Actin sense 5′-CTCTTCCAGCCTTCCTTCCT-3′

Actin antisense 5′-AGCACTGTGTTGGCGTACAG-3′

### 2.5. Cell Proliferation and Viability Assays

The CellTiter 96 AQueous One Solution Cell Proliferation Kit (Promega, Madison, WI, USA) was used for the 3-(4,5-dimethylthiazol-2-yl)-5-(3-carboxymethoxyphenyl)-2-(4-sulfophenyl)-2H-tetrazolium (MTS) assay to assess cell proliferation, according to the manufacturer’s protocol. The data were collected from quintuplicate measurements. For the cell viability assay, following transfection with SOX9 siRNA or miR-138pre, the treated cells were washed with phosphate-buffered saline. The Annexin V measurement was performed using the Muse Annexin V & Dead Cell Kit (Luminex, Austin, TX, USA), according to the manufacturer’s protocol.

### 2.6. Invasion Assay

In vitro invasion assays were performed using the Corning Biocoat Matrigel Invasion Chambers (Corning, Bedford, MA, USA), according to the manufacturer’s instructions. Briefly, the UC cell lines were seeded at a density of 2.5 × 10^4^ cells/well in a 24-well dish and transfected with 100 nmol/L SOX9 siRNA or miR-138pre for 72 h using Lipofectamine RNAiMAX (Life Technologies, Carlsbad, CA, USA). After culture, the samples were removed and re-plated in Matrigel chambers. After culturing for 72 h, the invading cells were stained and counted under a light microscope, according to the manufacturer’s instructions. The experiment was repeated thrice.

### 2.7. Statistical Analyses

Statistical analyses were conducted using the GraphPad Prism 8.0 software (GraphPad Software, Inc., La Jolla, CA, USA) with a two-tailed Student’s *t*-test to compare two groups. Graphical data are presented as the mean ± standard error of the mean. The results were considered statistically significant at *p* < 0.05. All experiments were performed with *n* ≥ 3.

## 3. Results

### 3.1. Evaluation of SOX9 Expression Levels in Non-Invasive and Invasive Urothelial Carcinoma via Immunohistochemical Staining

Immunohistochemical staining was performed to evaluate the expression of SOX9 protein in the carcinoma lesions of 67 urothelial carcinoma cases (pTa: 25 cases, pT1: 17 cases, pT2: 12 cases, and pTis: 13 cases) collected during TUR (Table 1, Figure 1). Intensity of the immunohistochemical staining was classified into 5 levels from 0 to 4. Although most non-invasive and non-muscle layer invasive urothelial carcinoma tissues showed negative or weakly positive intensity, SOX9 levels were significantly upregulated in muscle invasive urothelial carcinoma and carcinoma in situ (Figure 1).

### 3.2. SOX9 Expression Is Regulated by MiR-138 in Urothelial Carcinoma Cells

To analyze SOX9 function in vitro using urothelial carcinoma cell lines, SOX9 expression in urothelial carcinomas was first evaluated. Evaluation of SOX9 mRNA expression via quantitative RT-PCR showed high expression of SOX9 mRNA in T24 and UMUC2 cells (Figure 2A). This is consistent with the fact that SOX9 is highly expressed in invasive carcinomas in urothelial carcinoma tissue. Using these two cell types, we decided to analyze the function of SOX9 in urothelial carcinoma. To identify the miRNAs affecting SOX9 expression, we transfected urothelial carcinoma cell lines with various miRNA precursors (hsa-miR-138-5p, 675, 26a, 23b, 331-3p, 145, 27a, 345, 139, 197, and 367) that are important in different types of cancer and searched for miRNAs that significantly affected SOX9 mRNA expression. Several miRNAs suppressed SOX9 mRNA expression (Appendix A). Particularly, transfection with miR-138pre significantly reduced the SOX9 mRNA expression in both T24 and UMUC2 cells (Figure 2B). The introduction of the miR-138 precursor indicated that miR-138 regulates SOX9 expression. As evidence for this, we identified complementary sites on the mRNA sequence for the putative binding sites of miR-138. We found two putative binding sites in the Exon region and one in the 3’ untranslated region (Figure 2C and Appendix A). MTS assay was used to determine whether miR-138 affects the proliferative capacity of urothelial carcinoma cells. miR-138pre transfection significantly decreased the proliferation of T24 and UMUC2 cells (Figure 3A).

### 3.3. MiR-138 and SOX9 Are Involved in the Proliferation of T24, UMUC2, and UMUC3 Cell Lines

MTS assay revealed that transfection with miR-138pre decreased the proliferation of urothelial carcinoma cell lines. In addition, transfection with SOX9 siRNA significantly decreased the proliferation of T24, UMUC2, and UMUC3 cells. To elucidate the mechanism of cell proliferation in detail, we evaluated cell cycle, apoptosis, and autophagy. Transfection with miR-138pre or SOX9 siRNA resulted in G0/G1 cell cycle arrest (Figure 3B). Annexin V assay revealed the induction of early apoptosis (Figure 4A). Autophagy assay showed the induction of LC3 protein expression (Figure 4B).

### 3.4. MiR-138 and SOX9 Affect the Invasiveness of T24 and UMUC2 Cell Lines

Expression of SOX9 was significantly higher in invasive urothelial carcinoma cells than in non-invasive urothelial carcinoma cells, suggesting that miR-138 and SOX9 affect the invasive potential of these cells. To confirm this, invasion assays were performed using Matrigel. Inhibition of invasion was observed in both miR-138pre- and SOX9 siRNA-transfected cells using the Matrigel assay (Figure 5A). Furthermore, the correlation between increased SOX9 expression and vascular invasion in invasive urothelial carcinoma was histologically evaluated. A correlation between increased SOX9 expression and vascular invasion was observed in invasive carcinomas (Figure 5B). These results suggest that miR-138 and SOX9 contribute to the invasive potential of urothelial carcinoma cells. Notably, induction of miR-138 expression and suppression of SOX9 expression can suppress the invasive potential of urothelial carcinoma cells.

The increased expression of SOX9 in invasive carcinoma was also investigated at the mRNA level. At the same time, we also investigated the expression of miR-138 in the tissues. Tumor sections were excised from FFPE by macrodissection, RNA was extracted and SOX9 mRNA and miR-138 were quantified via quantitative PCR. Actin mRNA was quantified as an endogenous control, and samples from which good-quality RNA was extracted (pTa: 16, pT1:16, pT2:12, pTis:12) were compared based on pT stage, all of which were highly expressed in invasive cancer. The expression levels of SOX9 mRNA were consistent with the IHC results. Contrary to expectations, however, miR-138 showed the same trend as SOX9 mRNA.

## 4. Discussion

Here, we demonstrated that SOX9 plays an important role in the invasion of tumor cells at the cellular level, explaining the high expression of SOX9 in invasive urothelial carcinoma cells. Moreover, we identified miR-138 as a regulator of SOX9 expression in invasive urothelial carcinoma cells. miRNAs regulate the expression of numerous molecules that contribute to the growth and invasion of cancer cells, including urothelial carcinoma cells. miRNAs exert their tumor-specific effects by acting on several molecules, not just one target molecule. However, because miRNAs also maintain physiological functions that are essential for biological activities, their use in molecularly targeted therapies can trigger major disturbances in biological activities. Hence, the construction of an effective drug delivery system that targets only tumor cells is extremely difficult. In practice, deactivating or activating miRNAs for therapeutic purposes is difficult. Understanding the relationship between miRNAs and their target molecules will aid in the development of key molecules that can facilitate the diagnosis and treatment of cancer by clarifying the functions of miRNAs and their target molecules in cancer development and progression. In this study, we investigated the key molecules involved in the development and differentiation of urothelial carcinomas, explored their morphology, and searched for miRNAs regulating these molecules.

SOX9 is a member of the SOX transcription factor family [15] defined by a common HMG box domain originally identified in SRY, a sex-determining gene on the Y chromosome of SOX9 belonging to group E (SOX8, SOX9, and SOX10), which is involved in epithelial invasion, migration, and proliferation in prostate development and cancer [16,17]. SOX9 is involved in chondrocyte development as a master chondrogenic factor, and its expression is induced by receptor tyrosine kinase signaling [18]. In several carcinomas, SOX9 is also known to have important functions [19,20,21,22,23,24]. In urothelial carcinomas, SOX9 is significantly upregulated in invasive carcinomas and plays an important role in determining their invasive potential [25]. In this study, SOX9 expression levels were elevated in atypical invasive urothelial and intraepithelial carcinomas. In contrast, SOX9 expression was very low in low-grade non-invasive urothelial carcinomas with preexisting urothelial cell characteristics. High-grade invasive urothelial and intraepithelial carcinomas with high SOX9 expression were poorly differentiated tumors, suggesting that tumor cells have acquired diversity and are capable of various differentiations as a result of enhanced transcription factor function. Notably, miR-138 was identified as a regulator of SOX9 expression.

miR-138 is a tumor suppressor that targets various cancer-related genes. miR-138 acts as a post-transcriptional regulator of target molecules and inhibits cancer cell growth and invasion [26,27,28,29]. In hepatocellular carcinoma, it acts directly on cyclin D3, causing cell cycle arrest [1] and regulating the expression of SOX4, thereby controlling the growth of hepatocellular carcinoma [29]. In cholangiocarcinoma, miR-138 targets Bag-1 and inhibits cell proliferation [30]. In nasopharyngeal carcinoma, it inhibits tumor growth and development by regulating cyclin D1 expression [31]. It induces apoptosis in neuroblastoma [32] and regulates cancer growth in thyroid carcinoma and leukemia [33,34]. Furthermore, miR-138 contributes to the proliferation and invasion and plays important roles in the progression and prognosis of urological tumors, such as renal, urothelial, and prostate cancers [28,35,36,37].

In this study, miR-138 and its candidate target molecule, SOX9, were found to be involved in the proliferation of urothelial carcinoma cell lines. miR-138 overexpression suppressed the proliferation of urothelial carcinoma cells by enhancing early apoptosis and inducing autophagy. Matrigel assays revealed that miR-138 overexpression suppressed the invasion of urothelial carcinoma cells, similar to the suppression of SOX9 expression, suggesting that increased miR-138 expression and decreased SOX9 expression are critical for the growth and invasion of urothelial carcinoma. Low expression of SOX9 in non-invasive carcinoma was also consistent with the in vitro results in urothelial carcinoma tissues. The expression levels of SOX9 mRNA and miR-138 in urothelial carcinoma tissues were examined via quantitative RT-PCR analysis using RNA extracted from FFPE tissue samples. Although quantitative RT-PCR analysis using FFPE is simple and easy, and tumors can be easily sectioned through macrodissection, contamination of non-tumor areas may be a problem in small tissues such as carcinoma in situ. Quantification of miRNAs is further likely to be affected by non-tumor cells, such as inflammatory cells in the tumor, and should be interpreted with care, since miRNAs in the surrounding microenvironment as well as tumor cells may have been detected.

Urothelial carcinoma, also known as transitional epithelial carcinoma, exhibits diverse morphological changes, including squamous, glandular, and neuroendocrine differentiation. We previously showed that miR-145 induces the increased expression of stem cell markers and promotes squamous, glandular, and neuroendocrine differentiation [14]. Urothelial carcinoma cells are greatly affected by the expression of stem-cell-associated factors and transcription factors involved in early developmental differentiation, such as SOX9, which promote their differentiation and invasion (Figure 6).

## 5. Conclusions

In conclusion, we found that miR-138 regulated the growth and invasive potential of urothelial carcinoma cells by suppressing the expression of SOX9. Our results provide a basis for the development of effective strategies to modulate SOX9 expression using various drugs and molecules, including small RNAs such as miR-138.

## Figures and Tables

**Figure 1 biomedicines-11-03064-f001:**
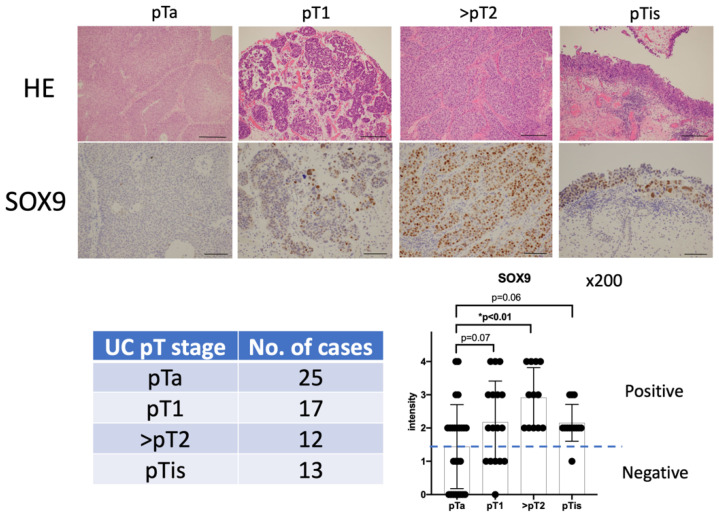
Expression levels of SRY-box transcription factor 9 (SOX9) in urothelial carcinoma tissues. (Upper panel) SOX9 expression levels in non-invasive and invasive urothelial carcinoma samples. The upper panel indicates hematoxylin–eosin stain (×100, scale bar:250 μm), and the lower panel indicates anti-SOX9 immunohistochemistry (×200, scale bar:100 μm). SOX9 expression at the nuclei was assessed semi-quantitatively as 0 (negative), 1 (weak intensity or <10% positive), 2 (intermediate intensity or 10–50% positive), 3 (strong intensity or 50–80% positive), and 4 (very strong intensity or >80% positive). (Lower left panel) Number of cases classified based on the pathologic T-stage of urothelial carcinoma samples. (Lower right panel) Distribution of the five-stage evaluation for each T stage of urothelial carcinoma, according to the staining intensity of immunohistochemical staining or percentage of SOX9-positive cells in the tumor.

**Figure 2 biomedicines-11-03064-f002:**
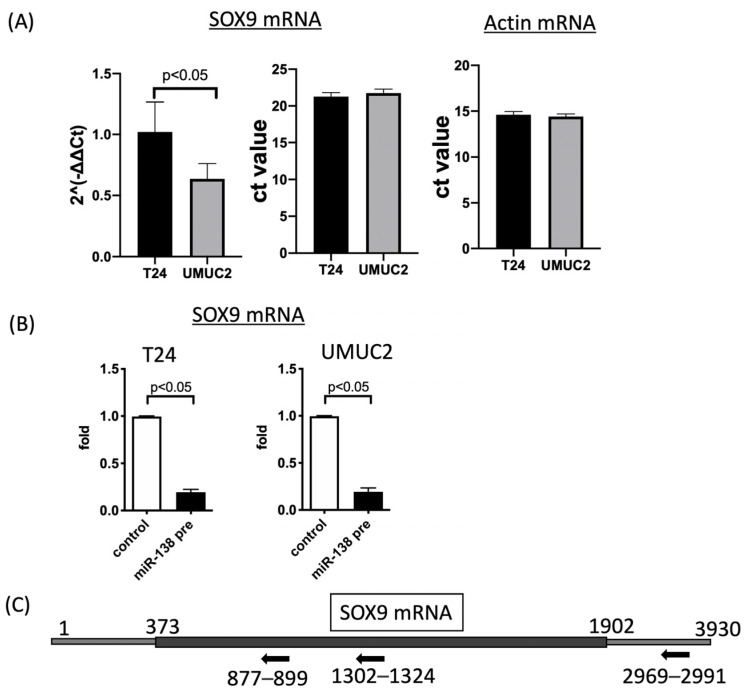
*SOX9* mRNA expression after transfection with various microRNA (miRNA) precursors. (**A**) Relative expression of *SOX9* mRNA after transfection of T24 and UMUC2 urothelial carcinoma cell lines with 11 miRNA precursors. (**B**) *SOX9* mRNA expression was significantly suppressed by miR-138pre transfection. (**C**) Predicted binding site of miR-138 in SOX9 mRNA. miR-138’s putative binding site is indicated by a black arrow (see Appendix A).

**Figure 3 biomedicines-11-03064-f003:**
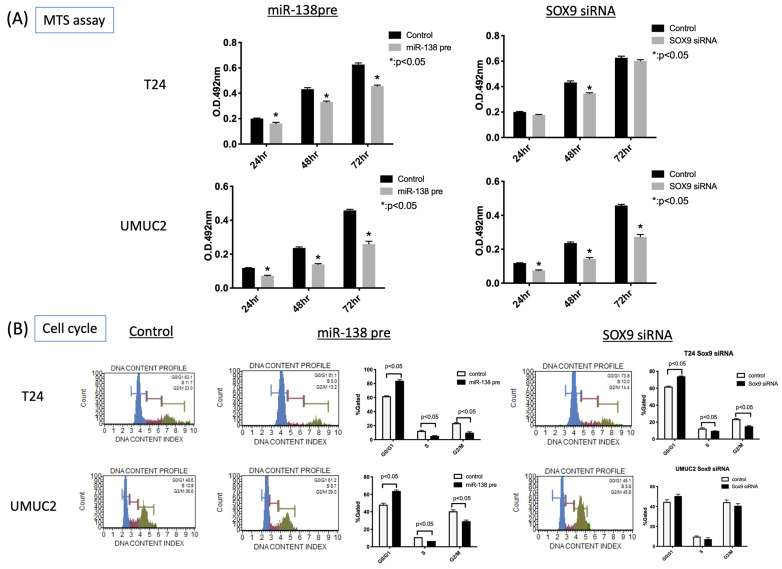
Effects of miR-138 and SOX9 on the proliferation of urothelial carcinoma cells. (**A**) 3-(4,5-dimethylthiazol-2-yl)-5-(3-carboxymethoxyphenyl)-2-(4-sulfophenyl)-2H-tetrazolium (MTS) assay after transfection of T24 and UMUC2 cells with *miR-138* or *SOX9* small interfering RNA (siRNA) at 24, 48, and 72 h. (**B**) Cell cycle analysis revealed that *miR-138pre* and *SOX9* siRNA transfection caused G1 arrest in the urothelial carcinoma cells.

**Figure 4 biomedicines-11-03064-f004:**
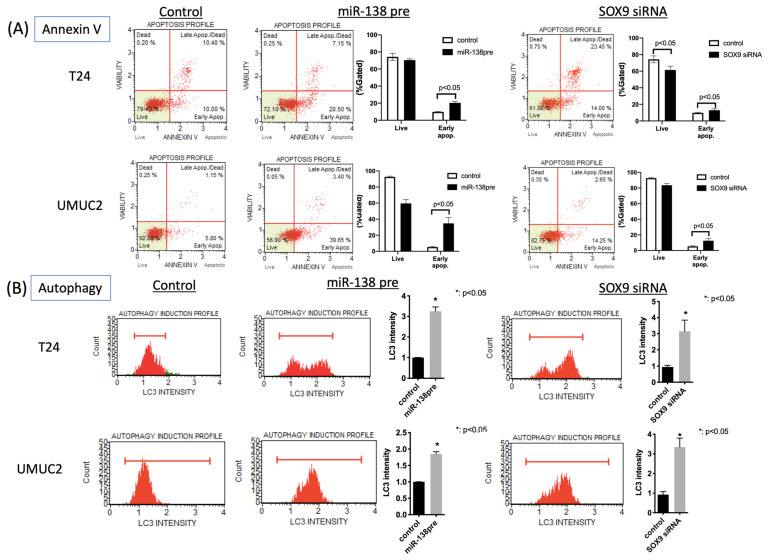
Functional analysis of *miR-138* overexpression or SOX9 suppression in T24 and UMUC2 cells. (**A**) Annexin V analysis of apoptosis in T24 and UMUC2 cells. (**B**) Autophagy analysis: changes in LC3 protein expression after miR-138pre or SOX9 siRNA transfection of T24 and UMUC2 cells.

**Figure 5 biomedicines-11-03064-f005:**
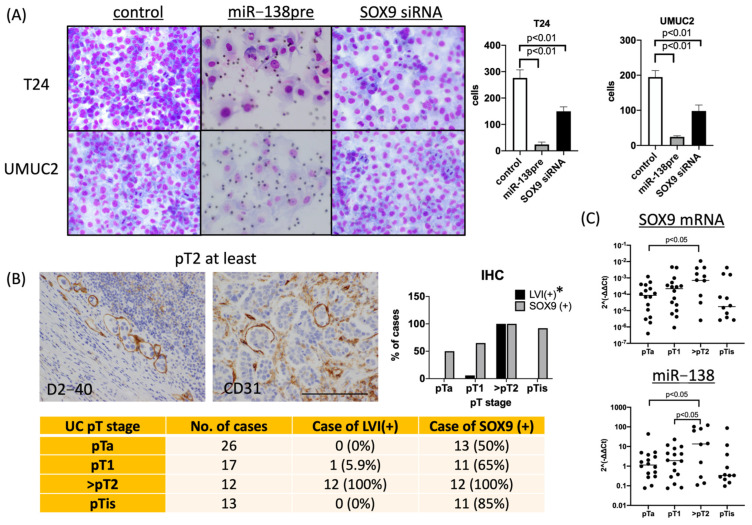
Evaluation of the invasive potential of urothelial carcinoma cells. (**A**) Matrigel assay using urothelial carcinoma cells. miR−138pre or SOX9 siRNA transfection suppressed the invasive potential of cells(×200). (**B**) Evaluation of vascular invasion in urothelial carcinoma tissues. * LVI+: Positive for lymphatic vessel invasion (×200). (**C**) Expression levels of SOX9 mRNA and miR−138 in tumor tissues at each pT stage.

**Figure 6 biomedicines-11-03064-f006:**
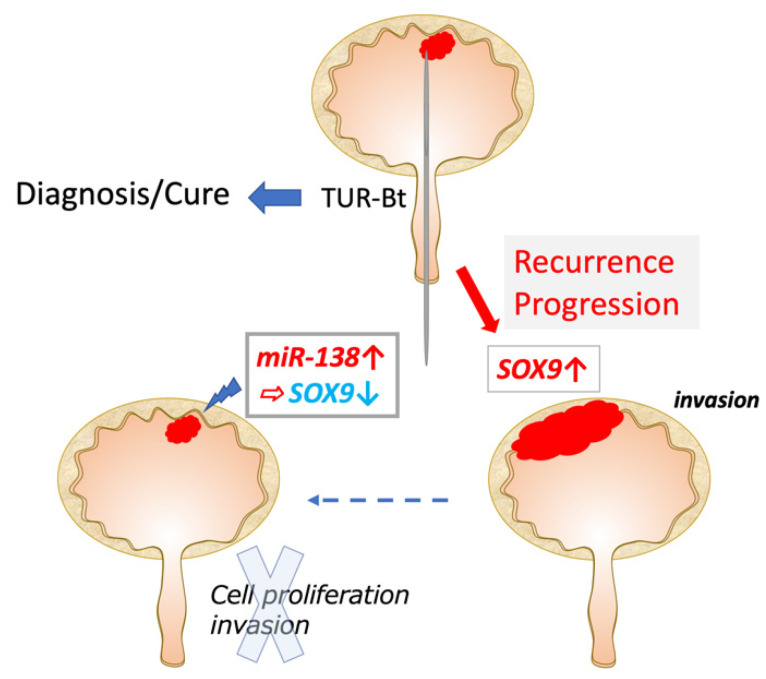
Function of SOX9 in UC. SOX9 are putative target molecules of miR-138. miR-138 and SOX9 regulate cell proliferative and invasive potential in UC.

**Table 1 biomedicines-11-03064-t001:** Patients’ Characteristics.

Number	67
Male	61
Female	5
Mean age (y.o.)	71.6 (37–95)
**pT stage of Urothelial carcinomas**	
pTa	25
pT1	17
≧pT2	12
pTis	13

## Data Availability

The data that support the findings of this study are available from the corresponding author upon reasonable request.

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
