# Peer review of "Overexpression of MicroRNA-138 Affects the Proliferation and Invasion of Urothelial Carcinoma Cells by Suppressing SOX9 Expression"

_biomedicines, 2023, doi:10.3390/biomedicines11113064_

Round 1

Reviewer 1 Report

Comments and Suggestions for Authors

Dear authors of “Overexpression of miRNA-138 affects the proliferation and invasion of urothelial carcinoma cells by suppressing SOX9 expression”,

Thanks for your contribution to this field. This is an interesting article aimed at determining the roles of SOX9 and miRNA-138 invasive urothelial carcinoma.

The manuscript is well written and organized. Research is well conducted. Obtained results are very convincing. The findings are of interest for urothelial cancer in general.

I may suggest some minor changes.

Lines 156-158 and Figure 2. Statistical analyses are required to estimate the effects of the various miRNA transfection on SOX9 mRNA. It is difficult to see the differences between panel A and B.

Figure 5, panel B: Please provide explanation about data collection, meaning of LVI+ and correct the mistake in the number of pTis case which should be 13 according to the numbers in Figure 1.

Looking forward seeing your modifications,

All the best,

Author Response

We sincerely thank you for reviewing our paper.

  1. Lines 156-158 and Figure 2. Statistical analyses are required to estimate the effects of the various miRNA transfection on SOX9 mRNA. It is difficult to see the differences between panel A and B.

Thank you for constructive reviewer’s comments.

As the reviewer points out, statistical analysis is necessary when comparing the effects of expression of several miRNAs. Since we derived Panel A as a pilot experiment, which provides the basis for the selection of miR-138, but not enough statistical evaluation, we present Panel A as supplementary data.

We added following description in Line 168-173.

“To analyze SOX9 function in vitro using urothelial carcinoma cell lines, SOX9 expression in urothelial carcinomas was first evaluated. Evaluation of SOX9 mRNA expression by quantitative RT-PCR showed high expression of SOX9 mRNA in T24 and UMUC2 cells. This is consistent with the fact that SOX9 is highly expressed in invasive carcinomas in urothelial carcinoma tissue. Using these two cell types, we decided to analyze the function of SOX9 in urothelial carcinoma.”

2.   Figure 5, panel B: Please provide explanation about data collection, meaning of LVI+ and correct the mistake in the number of pTis case which should be 13 according to the numbers in Figure 1.

We thank you for pointing out our carelessness.

In Figure 5, we have rewritten the correct number of cases and added the meaning of LVI+.

Reviewer 2 Report

Comments and Suggestions for Authors

This manuscript, entitled“Overexpression of MiRNA-138 Affects the Proliferation and Invasion of Urothelial Carcinoma Cells by Suppressing SOX9 3 Expression”, showed that miR-138–SOX9 signaling regulates the urothelial carcinomas proliferation and invasion. Obviously, the authors put a lot of effort into this manuscript, However, there are some major points that need to be addressed before publication:

1, Please provide detailed information about the patients.

2, Please add a scale to the diagram, such as in figure 1.

3, Is there a positive correlation between miR-138 and SOX9 in clinical patient samples?

4, How to prove that the SOX9 is the target of miR-138? Where is the binding site of miR-138?

5, Figure 6 looks too complex to understand.

Author Response

We sincerely thank you for reviewing our paper.

1, Please provide detailed information about the patients.

Thank you for constructive reviewer’s comments.

We provided information of patients in Table 1.

2, Please add a scale to the diagram, such as in figure 1.

Thank you for pointing out the lack of necessary information.

I have added a scale to each of the photos.

3, Is there a positive correlation between miR-138 and SOX9 in clinical patient samples?

Thanks for your very constructive feedback.

Tumor sections were excised from FFPE by macrodissection, and RNA was extracted and SOX9 mRNA and miR-138 were quantified by quantitative PCR.

Actin mRNA was quantified as an endogenous control, and samples from which good quality RNA was extracted (pTa:16, pT1:16, pT2:12, pTis:12) were compared by pT stage, all of which were highly expressed in invasive carcinoma.

The expression levels of SOX9 mRNA were consistent with the IHC results. Contrary to expectations, however, miR-138 showed the same trend as SOX9 mRNA.

Figure 5C was added to add this description in Line 231-238.

4, How to prove that the SOX9 is the target of miR-138? Where is the binding site of miR-138?

Thank you for constructive reviewer’s comments.

The introduction of the miR-138 precursor indicated that miR-138 regulates SOX9 expression. As evidence for this, we identified complementary sites on the mRNA sequence for the putative binding sites of miR-138. We found two putative binding sites in the Exon region and one in the 3' untranslated region (Figure 2C and Supplementary Figure 2).

This description was added in Lie 179-186.

5, Figure 6 looks too complex to understand.

We apologize for showing a complicated illustration.

We have simply shown what was revealed in this study.

Reviewer 3 Report

Comments and Suggestions for Authors

Dear authors,

I have reviewed your manuscript titled "Overexpression of MiRNA-138 Affects the Proliferation and Invasion of Urothelial Carcinoma Cells by Suppressing SOX9 Expression" and appreciate the opportunity to provide feedback. The study presents interesting findings on the role of miR-138 in regulating SOX9 expression and its impact on the proliferation and invasion of urothelial carcinoma cells. However, there are several areas that require attention to strengthen the scientific validity and clarity of the manuscript. I have outlined the major points below, and I encourage you to address them in your revisions.

1. In the results section, you transfected various miRNA precursors into the T24 and UMUC2 urothelial carcinoma cell lines to identify miRNAs that significantly affect SOX9 mRNA expression. The results demonstrate that transfection of miR-138 precursor significantly reduces SOX9 mRNA expression in T24 and UMUC2 cells, confirming the regulation of SOX9 expression by miR-138 in urothelial carcinoma cells. However, these findings are not sufficiently supported. Firstly, I recommend conducting qRT-PCR experiments to measure the expression levels of miR-138 and SOX9 in T24 and UMUC2 cells. This will provide quantitative data to support your observations. Secondly, I suggest performing experiments, such as dual-luciferase reporter assays, to directly demonstrate the ability of miR-138 to regulate SOX9 expression at the post-transcriptional level.

2. In the results section, you assessed the mechanisms underlying cell proliferation by evaluating cell cycle progression, apoptosis, and autophagy. While you provided detailed information on autophagy, it is important to include additional experiments to examine cell proliferation and apoptosis. I recommend conducting Western blot experiments to assess the expression of proliferation- and apoptosis-related proteins. This will provide a more comprehensive understanding of the mechanisms involved in cell proliferation.

3. The results of your study demonstrate that overexpression of miR-138 inhibits cell proliferation and invasion of urothelial carcinoma cells by suppressing SOX9 expression. However, the manuscript does not directly elucidate the interaction between miR-138 and SOX9. To address this gap, I suggest conducting rescue experiments. Specifically, you can include the following experimental groups: blank, miR-138 precursor, and miR-138 precursor + SOX9 vector. This will directly demonstrate whether overexpression of SOX9 can reverse the inhibitory effects of miR-138 on cell proliferation and invasion in urothelial carcinoma cells.

4. In Figure 4, the figure caption mentions TJP2, but this term is not mentioned elsewhere in the manuscript. Please verify whether this is an error or if there is a missing reference to TJP2 in the text. Ensure consistency throughout the manuscript.

Addressing these points will significantly enhance the scientific validity and clarity of your manuscript. I encourage you to carefully consider these suggestions and make the necessary revisions accordingly. Once these revisions are made, I would be happy to re-evaluate the manuscript.

Thank you for your contribution to the field, and I look forward to seeing the revised version of your manuscript.

Comments on the Quality of English Language

The author's manuscript is well written and of a high standard. However, there are a few sentences that need further improvement. I believe that with some fine-tuning, this production will reach its full potential.

Author Response

We sincerely thank you for reviewing our paper.

  1. In the results section, you transfected various miRNA precursors into the T24 and UMUC2 urothelial carcinoma cell lines to identify miRNAs that significantly affect SOX9 mRNA expression. The results demonstrate that transfection of miR-138 precursor significantly reduces SOX9 mRNA expression in T24 and UMUC2 cells, confirming the regulation of SOX9 expression by miR-138 in urothelial carcinoma cells. However, these findings are not sufficiently supported. Firstly, I recommend conducting qRT-PCR experiments to measure the expression levels of miR-138 and SOX9 in T24 and UMUC2 cells. This will provide quantitative data to support your observations. Secondly, I suggest performing experiments, such as dual-luciferase reporter assays, to directly demonstrate the ability of miR-138 to regulate SOX9 expression at the post-transcriptional level.

Thank you for your very constructive comments.

The T24 and UMUC2 used in this study have high expression of SOX9 mRNA, consistent with the high expression of SOX9 protein and mRNA in invasive urothelial carcinoma, as shown in the tissue.

We have added this to Figures 2Aand 5C. Confirmation of the putative binding site confirmed the presence of a homologous binding site for miR138 on the mRNA sequence. This was noted in Figure 2C, supplemental Figure 2 and lines 179 to 186.

  1. In the results section, you assessed the mechanisms underlying cell proliferation by evaluating cell cycle progression, apoptosis, and autophagy. While you provided detailed information on autophagy, it is important to include additional experiments to examine cell proliferation and apoptosis. I recommend conducting Western blot experiments to assess the expression of proliferation- and apoptosis-related proteins. This will provide a more comprehensive understanding of the mechanisms involved in cell proliferation.

Thanks for your very constructive comments.

As the reviewer stated, confirming the expression of molecules related to the mechanism of proliferative capacity would be useful supportive data for the present results. However, 10 days is too short of a deadline to revise the results in order to carry out these studies.

If allowed, we would like to have 1.5 months more time.

As an additional experiment, we believe that proteins to be detected by Western blotting, such as bcl2, BAX, and BclXL for apoptosis, and LC3 and p62 for autophagy, would be useful.

  1. The results of your study demonstrate that overexpression of miR-138 inhibits cell proliferation and invasion of urothelial carcinoma cells by suppressing SOX9 expression. However, the manuscript does not directly elucidate the interaction between miR-138 and SOX9. To address this gap, I suggest conducting rescue experiments. Specifically, you can include the following experimental groups: blank, miR-138 precursor, and miR-138 precursor + SOX9 vector. This will directly demonstrate whether overexpression of SOX9 can reverse the inhibitory effects of miR-138 on cell proliferation and invasion in urothelial carcinoma cells.

Thanks for your very constructive comments.

As the reviewer mentioned, the experiment to see if we can exogenously overexpress SOX9 and suppress it with miR-138 would be very useful and would provide supporting data to confirm the present study. This study also partially supports that SOX9 is a target molecule of miR-138. We would very much like to add this study, but since it is difficult to do so within the given 10-day REVISE period, we would like to ask for an extension of another 1.5 months.

  1. In Figure 4, the figure caption mentions TJP2, but this term is not mentioned elsewhere in the manuscript. Please verify whether this is an error or if there is a missing reference to TJP2 in the text. Ensure consistency throughout the manuscript.

We thank you for pointing out our carelessness.

We have rewritten TJP2 to SOX9.

Round 2

Reviewer 2 Report

Comments and Suggestions for Authors

I suggest the authors do luciferase reporter assays to directly demonstrate that sox9 is a direct target for miRNA. And discuss why the “miR-138 showed the same trend as SOX9 mRNA” in clinical patient samples. 

Author Response

We thank you for the opportunity to resubmit our manuscript to the Applied Sciences.

I suggest the authors do luciferase reporter assays to directly demonstrate that sox9 is a direct target for miRNA. And discuss why the “miR-138 showed the same trend as SOX9 mRNA” in clinical patient samples.

Thank you for constructive reviewer’s comments.

Reporter assays are an important consideration in this study. Additional experiments will be performed according to its advice.

To do this, we will start with the construction of the reporter vector construct. This will take about 2 months of your time, which is far beyond our 5-day revise period, so we request that you give us 2 months for the additional experiments.

As the reviewer points out, the fact that miR-138 and SOX9 show the same trend in cancer tissue is inconsistent with our hypothesis.

We do not know the real reason for this. However, we can assume the following;

The expression levels of SOX9 mRNA and miR-138 in urothelial carcinoma tissues were examined by quantitative RT-PCR analysis using RNA extracted from FFPE tissue samples. Although quantitative RT-PCR analysis using FFPE is simple and easy, and tumors can be easily sectioned by macrodissection, contamination of non-tumor areas may be a problem in small tissues such as carcinoma in situ. Quantification of miRNAs is further likely to be affected by non-tumor cells, such as inflammatory cells in the tumor, and should be interpreted with care, since miRNAs in the surrounding microenvironment as well as tumor cells may have been detected.

We have included this statement in Discussion.(Line 305-312)

Reviewer 3 Report

Comments and Suggestions for Authors

Dear Authors,

I hope this letter finds you well. I have carefully reviewed your manuscript titled “Overexpression of MiRNA-138 Affects the Proliferation and Invasion of Urothelial Carcinoma Cells by Suppressing SOX9 Expression”, and I appreciate the effort you have put into addressing the concerns raised during the initial review. I am pleased to inform you that I am willing to grant an extension of 1.5 months for you to complete the revisions.

1. Regarding the first issue you mentioned, your explanation and response are reasonable and satisfactory. I find your clarification acceptable, and there is no further action required on this matter.

2. With respect to the second issue, I fully support your suggestion to conduct additional experiments to strengthen the findings. The inclusion of these experiments will undoubtedly enhance the overall quality and impact of your manuscript. I also acknowledge your request for an extension of 1.5 months to complete these experiments, and I am pleased to grant you the additional time.

3. Concerning the third issue, I agree with your request for an extension of 1.5 months to address the concerns raised. This additional time will allow you to thoroughly revise the relevant sections and ensure that all necessary changes are implemented appropriately.

4. Lastly, I have carefully reviewed the revised version of your manuscript and have taken note of the modifications you made in response to the fourth issue. It is evident that you have taken the reviewer's comments into consideration, and the changes made have improved the clarity and overall quality of the manuscript.

In summary, I am pleased with the progress you have made in addressing the concerns raised during the initial review. I am willing to grant an extension of 1.5 months to allow you to complete the necessary revisions and experiments. Please ensure that you carefully consider the reviewers' comments and suggestions throughout the revision process. I encourage you to pay particular attention to any additional experiments and provide a thorough explanation of the results obtained.

Thank you for your dedication and effort in improving the manuscript. I look forward to receiving your revised version within the specified time frame. Should you have any questions or require further clarification, please do not hesitate to contact me.

Author Response

We thank you for the opportunity to resubmit our manuscript to the Applied Sciences.

  1. Regarding the first issue you mentioned, your explanation and response are reasonable and satisfactory. I find your clarification acceptable, and there is no further action required on this matter.

Thank you for appreciating our reconsideration.

  1. With respect to the second issue, I fully support your suggestion to conduct additional experiments to strengthen the findings. The inclusion of these experiments will undoubtedly enhance the overall quality and impact of your manuscript. I also acknowledge your request for an extension of 1.5 months to complete these experiments, and I am pleased to grant you the additional time.

Thank you for giving me the time to conduct additional experiments. We will conduct additional experiments and add them to this manuscript as soon as possible.

  1. Concerning the third issue, I agree with your request for an extension of 1.5 months to address the concerns raised. This additional time will allow you to thoroughly revise the relevant sections and ensure that all necessary changes are implemented appropriately.

Thank you for giving me the time to conduct additional experiments. We will conduct additional experiments and add them to this manuscript as soon as possible.

  1. Lastly, I have carefully reviewed the revised version of your manuscript and have taken note of the modifications you made in response to the fourth issue. It is evident that you have taken the reviewer's comments into consideration, and the changes made have improved the clarity and overall quality of the manuscript.

Thank you for agreeing with our discussion. We appreciate your comments.